# Modulation of Multispecific Transporters by *Uncaria tomentosa* Extract and Its Major Phytoconstituents

**DOI:** 10.3390/pharmaceutics16111363

**Published:** 2024-10-25

**Authors:** Nóra Szilvásy, Panna Lajer, Attila Horváth, Katalin Veres, Judit Hohmann, Zsuzsanna Schelz, Renáta Minorics, István Zupkó, Zsuzsanna Gáborik, Emese Kis, Csilla Temesszentandrási-Ambrus

**Affiliations:** 1Charles River Laboratories Hungary, H-1117 Budapest, Hungary; nora.szilvasy@crl.com (N.S.); panna.lajer@gmail.com (P.L.); zsuzsanna.gaborik@crl.com (Z.G.); emese.kis@crl.com (E.K.); 2Institute of Pharmacognosy, University of Szeged, Eötvös Street 6, 6720 Szeged, Hungary; sztewee@gmail.com (A.H.); veres.katalin@szte.hu (K.V.); hohmann.judit@szte.hu (J.H.); 3ELKH-USZ Biologically Active Natural Products Research Group, University of Szeged, 6720 Szeged, Hungary; 4Institute of Pharmacodynamics and Biopharmacy, University of Szeged, Eötvös Street 6, 6720 Szeged, Hungary; schelz.zsuzsanna@szte.hu (Z.S.); kanizsaine.minorics.renata@szte.hu (R.M.); zupko.istvan@szte.hu (I.Z.)

**Keywords:** SLC, ABC, transporter, *Uncaria tomentosa*, herb–drug interaction, in vitro

## Abstract

**Background/Objectives:** One of the major risks associated with the concomitant use of herbal products and therapeutic drugs is herb–drug interactions (HDIs). The most common mechanism leading to HDIs is the inhibition and/or induction of transport proteins and drug-metabolizing enzymes by herbal ingredients, causing changes in the pharmacokinetic disposition of the victim drug. The present study aimed to determine the potential interactions of *Uncaria tomentosa* (UT) (cat’s claw), a popular herb due to its supposed health benefits. **Methods**: The effect of UT extract and its major oxindole alkaloids was investigated on multispecific solute carrier (SLC) and ATP-binding cassette (ABC) drug transporters, using SLC transporter-overexpressing cell lines and vesicles prepared from ABC transporter-overexpressing cells. **Results**: UT extract significantly inhibited all ABC transporters and the majority of the SLC transporters tested. Of the investigated oxindole alkaloids, isopteropodine significantly inhibited OATP, OCT1 and OCT2, OAT3, ENT4, MDR1, and BCRP transporters. OCTs, OCTN1-, ENT1-, and MDR1-mediated substrate accumulation was below 50% in the presence of mitraphylline. **Conclusions**: Based on the calculated intestinal concentration of UT extract, interactions with intestinal transporters, especially OATP2B1, ENTs, MRP1, MRP2, MDR1, and BCRP could be relevant in vivo. Our data can help to predict the clinical consequences of UT co-administration with drugs, such as increased toxicity or altered efficacy. In conclusion, the use of these in vitro models is applicable for the analysis of transporter-mediated HDIs similar to drug–drug interaction (DDI) prediction.

## 1. Introduction

The use of over-the-counter products with bioactive ingredients for medicinal purposes is increasing worldwide and may be the source of numerous clinically relevant and sometimes life-threatening herb–drug interactions that are not anticipated in the routine clinical setting [1]. It is widely assumed that herbal preparations are devoid of side effects and are safe to consume; however, herb–drug interactions (HDIs) are clearly not negligible [2,3]. HDIs can alter the pharmacokinetics or pharmacodynamics of a drug by changing the absorption, distribution, metabolism, excretion, and toxicity (ADMET) properties as a result of metabolizing enzymes and drug transporter modulation [4]. The widely used herb, *Uncaria tomentosa* (Willd. ex Schult.) DC. (UT), a prominent member of the *Uncaria* genus in the Rubiaceae family and a curved, hooked, and thorned creeper plant commonly referred to as cat’s claw, is native to the Amazon rainforest and other tropical areas of South and Central America. The healing effects of UT are used by Peruvian tribes to treat a multitude of diseases including inflammations, cancer, gastric ulcers, arthritis, and infections [5,6,7,8]. Consequently, in vitro and pharmacological studies of UT mainly focus on its potential immunomodulatory and anti-inflammatory effects in diseases such as osteoarthritis, asthma, and arthritic joint disease [9,10,11,12,13,14,15,16,17,18,19,20,21,22], and its cytotoxic, apoptotic, and antiproliferative effects on various types of cancer [23,24,25,26,27,28,29,30,31,32,33,34]. In addition, central nervous system (CNS)-related effects were also investigated [35,36]. In clinical trials involving patients with rheumatoid arthritis, a reduction in the number of painful and swollen joints was observed [37]. However, only weak evidence was found in a clinical trial aiming to support the immunostimulant activity of UT extract after pneumococcal vaccination [38,39]. To date, neither the U.S. Food and Drug Administration (FDA) have approved cat’s claw for medicinal use, nor has the European Medicines Agency (EMA) established a herbal monograph for UT; therefore, it is commercially available as a dietary supplement and not as an herbal medicinal product [40,41]. Up to now, approximately 50 compounds have been isolated from UT extracts and classified as alkaloids, flavonoids, and phenolic compounds [42]. Phytochemical investigations revealed the presence of oxindole and indole alkaloids, organic acids, polyphenols, triterpenes, and sterols as major classes of compounds in UT. The pharmacological effects of the bark extract are attributed to the tetracyclic (TOAs) and pentacyclic oxindole alkaloids (POAs), including uncarine D, mitraphylline, isomitraphylline, pteropodine, isopteropodine, and uncarine F [43,44]. Concerning the safety of the clinical application of UT extract, there are only sparse published HDI data for UT. It has been demonstrated that UT extract significantly induced the mRNA expression of CYP3A4, CYP2J2, UGT1A3, UGT1A9, MDR1, and OATP1B1, and strongly activated the pregnane X receptor (PXR) and aryl hydrocarbon receptor [45,46]. Seven alkaloids (isocorynoxeine, rhynchophylline, isorhynchophylline, corynoxeine, isopteropodine, pteropodine, and mitraphylline) from cat’s claw extracts were identified as PXR activators. Beyond this, limited information is available on the transporter interaction with UT extract and its biologically active ingredients [36,45].

Drug transporters are members of either of two major superfamilies, ABC and SLC. These are transmembrane proteins widely expressed in the epithelial cells of various tissues with especially high expression in organs involved with either absorption or excretion (the liver, kidney, and intestine) or with barrier functions (the brain, placenta, and retina). The investigation of drug transporters may lead to a better understanding of how they affect a drug’s oral bioavailability, toxicity, efficacy, tissue distribution, organ-specific entry, and DDIs and HDIs (Appendix A). Approximately 60% of the commercially available small molecule pharmaceuticals are orally administered [47] and are absorbed from the intestinal epithelium. One of the four pharmacokinetic phases primarily affecting drug plasma concentration is intestinal absorption. In the intestine, apical MDR1, MRP2, and BCRP extrude their substrates back to the lumen, decreasing their bioavailability, while uptake transporters including OCTs, LATs, OCTN1, ENT1, ENT2, ENT4, THTR2, and OATP2B1 facilitate the penetration of their substrates into the cells. The basolateral efflux transporters MRP1, MRP3, and MRP4 mediate the transport into the portal vein. Drug transporters in the liver can be regarded as completing the phase I and II enzyme-based detoxification process; drug uptake mediated by OAT2, OCT1, OCT3, OCTN1, NTCP, ENT1, ENT2, and OATPs (1B1, 1B3, 2B1) delivers the drug to the detoxification system to facilitate metabolism, whereas drug efflux by MDR1, BCRP, BSEP, MRP2, and MATE1 decreases the load on detoxification enzymes. Some of the liver conjugates are then excreted as bile components back into the intestine, while the basolateral efflux pumps MRP1/3/4/5, but not MRP2, transport their substrates into the blood, allowing them to reach the systemic circulation. Several drugs cause cholestasis by inhibiting canalicular efflux transporters [48]; however, there are significant differences in the relevance of biliary transporter inhibition in the development of drug-induced liver injury (DILI) [49]. In the systemic circulation, the compounds reach the organs and blood–tissue barriers such as the blood–brain barrier (BBB), where the apically localized MDR1, BCRP, MRP2, and MRP4 with their broad substrate specificity restrict the penetration of drugs into the CNS while THTR2 and OATPs including 2B1 and 1A2 can facilitate drug transport across the BBB [50,51]. The kidney, the main excretory organ, has key roles in the reabsorption and excretion of compounds. Here, the renal transporters of SGLT2, THTR2, ENT2, URAT1, MRP1, MRP3, and MRP5 mediate the reabsorption of their substrates from the urine while OCTs, OATs, ENT1, MDR1, MRP2, MRP4, MATE1, MATE2K, THTR1, and BCRP contribute to the excretion of the compounds from the body, working in concert on the basolateral and apical sides of proximal tubule cells [52,53,54,55,56]. An imbalance between transporter-mediated uptake and efflux may result in drug accumulation in proximal tubule cells, leading to drug-induced nephrotoxicity and kidney injury [57,58]. In summary, UT, as with any herb, has a multitude of mechanistic options to affect ADMET. Despite the widely known safety concerns of drug–food interactions, the drug development process is still lacking robust methodological guidelines that could help create definitive clinical and regulatory recommendations. This study assesses the potential for a herb–drug interaction of UT extract and its selected bioactive compounds, mitraphylline, uncarine D, and isopteropodine. We examined their intrinsic properties as a perpetrator to interfere with drug absorption and disposition via SLC and ABC transporters. The results were also used to predict the potential for clinical pharmacokinetic interference between UT and co-administered drugs.

## 2. Materials and Methods

### 2.1. Compounds

Reagents and non-radiolabeled chemicals were purchased from Merck KGaA (Darmstadt, Germany). All chemicals were of analytical grade. Tritium-labeled estrone-3-sulfate (^3^H-E3S) and N-methyl-quinidine (^3^H-NMQ) were purchased from the Biological Research Centre (Szeged, Hungary). The tritiated cholecystokinin octapeptide (^3^H-CCK-8), 1-methyl-4-phenylpyridinium (^3^H-MPP^+^), dehydroepiandrosterone sulfate (^3^H-DHEAS), estradiol 17β-D-glucuronide (^3^H-E_2_17βG), taurocholic acid (^3^H-TC), and Ultima Gold XR scintillation fluid were purchased from PerkinElmer (Waltham, MA, USA). ^14^C-metformin, ^14^C methyl α-D-glucopyranoside (^14^C-AMG), and ^3^H-tenofovir were from American Radiolabeled Chemicals, Inc. (St. Louis, MO, USA) and Moravek Inc. (Brea, CA, USA). Hanks’ Balanced Salt Solution (HBSS) (10×) was purchased from Life Technologies (Waltham, MA, USA). Mitraphylline (purity 99.66%) and uncarine D (purity 97%) were from PhytoLab GmbH & Co. KG (Vestenbergsgreuth, Germany), and isopteropodine (USP reference standard) was obtained from Merck KGaA.

### 2.2. Chemicals and Reagents

Methanol used for plant extraction was analytical grade (Molar Chemicals Kft, Halásztelek, Hungary). Purified and HPLC grade water was obtained with a Millipore Direct-QVR 3 UV pump (Millipore S. A. S., Molsheim, France). Acetonitrile and methanol (MeOH) used for sample preparation and HPLC were HPLC grade and purchased from Merck KGaA.

### 2.3. Plant Material

Inner bark of the stem of *Uncaria tomentosa* (Willd. ex Schult.) DC. was purchased from Manu JTC, Budapest, Hungary. The plant material originates from Peru. A small amount of chopped bark was preserved as voucher specimens (No. 896) in the Institute of Pharmacognosy, Szeged, Hungary.

### 2.4. Preparation of the Plant Extract (UT Extract)

200.7 g ground plant material was extracted with 12 L 90% MeOH by percolation at room temperature. The extract was concentrated by Rotavapor in vacuum to 50 mL, and then made to dry by lyophilization using a Christ Alpha 1-4 LSCplus (Osterode am Harz, Germany) instrument. The yield was 7.21% (14.48 g).

### 2.5. Quantitative Analysis of the Extract

*Instrumentation*: Liquid chromatographic analyses of UT extract were performed using a Shimadzu (Shimadzu Corporation, Kyoto, Japan) system equipped with an LC-20 AD pump, a DGU-20A_SR_ degasser unit, an SPD-M20A Diode Array detector, a CBM-20A Controller, and a SIL-20A_HT_ Autosampler. Data assessment was performed with the LabSolutions (Version 5.82) software (Shimadzu, Kyoto, Japan). Chromatographic separations were developed on a Gemini-NX column (C-18 110A 3 µm, 150 × 4.60 mm) under a gradient program system by changing the ratio of acetonitrile (MeCN) in H_2_O (containing 50 mM of NH_4_-formiate) as follows: 40% (0−1 min), 40–45% (1−10 min), 45–90% (10−15 min), keeping it at 90% for 1 min, 90% to 40% (16−17 min), and keeping it at 40% for 3 min. The flow rate was set at 1.2 mL/min, the injection volume was 10 μL, and the analyte was monitored in the UV–Vis range (190–800 nm) and at UV_max_ 241 nm of the standards (mitraphylline, uncarine D, and isopteropodine). The temperature was set to 25 °C.

*Validation*: Major components of UT, mitraphylline, uncarine D, and isopteropodine as standards were used to analyze the composition of the extract. The retention time of isopteropodine was 9.2 min, while that of mitraphylline and uncarine D was the same (4.6 min) (Appendix A). Therefore, the sum of mitraphylline and uncarine D was determined and expressed as mitraphylline. Calibration curves and limit of detection (LoD) and limit of quantitation (LoQ) values (Appendix A) were established for mitraphylline and isopteropodine. Calibration curves were determined based on 10 calibration points. The correlation coefficient of the calibration curves was at least 0.9999.

*Sample preparation*: A total of 100 mg of lyophilized 90% MeOH extracts of *U. tomentosa* was in a 10 mL volumetric flask and dissolved in MeOH aided by an ultrasonic bath. After homogenization, the sample was filtered by a PTFE 0.45 μm filter (FilterBio PTFE-L Syringe filter, Labex Ltd., Budapest, Hungary) and the first 1 mL was discarded. From the extract, three sample preparations were made, and each sample was injected in triplicate. All extracts were stored in the refrigerator until analysis. The composition of the dried extract of UT—free from TOA—prepared from the ground inner bark with 90% MeOH was characterized by the HPLC-DAD method. The isopteropodine content in the extract was determined to be 4.265 ± 0.1936 mg/g, and the mitraphylline + uncarine D content to be 9.430 ± 0.6407 mg/g and expressed as mitraphylline.

### 2.6. Cell Culture

The human embryonic kidney 293 (HEK293) cell line was purchased from Invitrogen/ThermoFisher (Waltham, MA, USA). Madin–Darby canine kidney II (MDCKII) wild type cells were obtained from the European Collection of Authenticated Cell Cultures (ECACC catalog no. 00062107). HEK293 cells overexpressing the human uptake OATP1B1 (NCBI Reference Sequence: NM_006446.4), OATP1B3 (NM_019844), OATP2B1 (NM_007256), OATP1A2 (NM_134431), OAT1-3 (BC033682.1; NM_006672.3; BC022387), OCT1-3 (BC126364.1; NM_003058; NM_021977), OCTN1 (NM_003059.3), NTCP (NM_003049.3), MATE1 (NM_018242.3), MATE2K (NM_001099646.2), ENT4 (NM_001040661.1), SGLT2 (AJ133127.1), ASCT1/2 (NM_003038.5; NM_005628.3), LAT1/2 (NM_003486.6; NM_012244.4), and THTR1/2 (NM_006996; NM_025243) and mock and human efflux BCRP (NM_004827.2), BSEP (NM_003742.2), MDR1 (NM_000927.4), and MRP1-5 (NM_004996; NM_000392.4; BC137347.1; NM_005845.4; NM_005688.2) transporters as well as MDCKII overexpressing human ENT1/2 (NM_001078177; NM_001532.2) and URAT1 (BC053348) transporters were created by lentiviral transduction. The cDNA encoding respective transporters were synthetized and cloned by GenScript (Piscataway, NJ, USA). The production of lentivirus supernatants, the transduction of the cells, cloning, functional characterization, and validation were performed by Charles River Laboratories Hungary Kft. (Budapest, Hungary).

Cell cultures were maintained in Dulbecco’s modified Eagle’s medium (DMEM), 4.5 g/L of glucose, supplemented with GlutaMAX^TM^, 2.5 *v*/*v* % penicillin/streptomycin, and 10% fetal bovine serum (FBS) at 37 °C, 5% CO_2,_ and 90% relative humidity. The medium was replaced two or three times per week; cells were harvested using TrypLE™ Express (ThermoFisher) at 80 to 90% confluence and passaged or seeded. HEK293 cells of passage < 12 (from thawing) and MDCKII cells of passage < 18 were seeded at a density of 1 × 10^5^ cells/well onto poly-D-lysine-coated or -non-coated 96-well plates and incubated for 18–22 h prior to inhibition assays.

### 2.7. Membrane Vesicle Preparation

Membrane vesicles were prepared from ABC transporter-overexpressing HEK293 cells by Charles River Laboratories Hungary Kft. as described previously in detail [59]. Vesicles were frozen in liquid nitrogen and stored at −80 °C.

### 2.8. Uptake Inhibition Assays

Plated cells were preincubated with a transport buffer (HBSS) containing 30 or 150 µg/mL of UT extract or 100 µM of isopteropodine, mitraphylline, or uncarine D for 15 min at an appropriate temperature. For IC_50_ determination, half dilution series were used from a maximum concentration of 150 µg/mL (UT extract) or from 100 µM (isopteropodine, mitraphylline, or uncarine D). Uptake inhibition experiments were initiated by replacing the buffer solution with a dosing solution containing the appropriate substrate and the UT extract or oxindole alkaloids for a previously validated incubation time (Appendix A). Uptake was terminated at the indicated time points by aspirating the dosing solution, and cells were washed with an ice-cold assay buffer and lysed in 0.1 M NaOH. For radiolabeled substrates, accumulation was measured by liquid scintillation counting (MicroBeta Scintillation Counter, PerkinElmer).

### 2.9. Vesicular Transport Assays

Vesicular transport inhibition assays were performed using a rapid filtration technique [60]. In brief, the appropriate amount of the protein of each membrane vesicle was mixed with either the extract or isopteropodine, mitraphylline, or uncarine D at one concentration or with their half dilution series from 150 µg/mL or from 100 µM, the suitable transporter substrate, and the assay buffer on ice. Reaction mixtures were preincubated for 15 min at 37 °C with shaking at 250 rpm. Following preincubation, 4 mM Mg-ATP or AMP was added and incubated according to Appendix A. Vesicular transport was terminated by adding an ice-cold washing buffer, and the reaction mix was transferred to a MSFBN6B10 filter plate (Merck) for filtration. The filter plates were then dried at 40 °C for 4 h, and radiolabeled substrates retained in the vesicles were detected by liquid scintillation counting as described above. For fluorescent substrates, accumulation was measured with a Clariostar^plus^ microplate reader (BMG Labtech GmbH, Ortenberg, Germany).

### 2.10. Antiproliferative Assays

For the determination of the antiproliferative properties of the prepared extract and selected alkaloids (isopteropodine, mitraphylline, and uncarine D), an MTT assay was applied on our HEK293 cell panel (see Appendix A) [61]. A panel of human adherent cancer cell lines including breast (T47D, MCF7, and MDA-MB-231), cervix (HeLa, SiHa, and C33A), ovarian (A2780), and murine fibroblast (NIH/3T3) cells was additionally utilized to study the effect of the prepared extract, selected alkaloids (isopteropodine, mitraphylline, and uncarine D), and cisplatin on cell growth (Appendix A). All were maintained in a minimal essential medium completed with 10% fetal bovine serum, 1% non-essential amino acids, and 1% penicillin–streptomycin at 37 °C in a humidified atmosphere containing 5% CO_2_. All cell lines were purchased from the European Collection of Cell Cultures (Salisbury, UK), except for SiHa and C33A, which were from the American Tissue Culture Collection, LGC Standards GmbH (Wesel, Germany). Cell culture media and supplements were obtained from Lonza Group Ltd. (Basel, Switzerland). Cells were seeded into 96-well plates (C33A 10,000/well, all others 5000/well); after overnight incubation, the test substances were added and incubated for another 72 h under cell-culturing conditions. *Uncaria tomentosa* extract and the alkaloids were tested in the 1–90 μg/mL and 1–30 μM ranges, respectively. Finally, 20 μL of 5 mg/mL MTT solution was added to each well and incubated for 4 h; the medium was discarded, and the precipitated formazan crystals were dissolved in DMSO for 60 min with shaking. The absorbance was determined using a microplate reader (SPECTROstar Nano, BMG Labtech GmbH). Two independent experiments were performed with five parallels for each condition. Cisplatin, a clinically used anticancer agent (Ebewe GmbH, Unterach, Austria), was included as a reference compound. Calculations were conducted using the GraphPad Prism 10.0 software (GraphPad Software Inc., San Diego, CA, USA).

### 2.11. Calculations

I_gut_ values were calculated as follows [62]:(1)$$\frac{dose}{250\;mL}$$
where I_gut_ is the estimated gut concentration of the extract or its components and dose is the applied amount of the administered UT extract. Considering the wide variety of recommended doses and products, a 500 mg dose was used for the calculations here. I_gut_/IC_50_ ratios were calculated for the inhibitors of MDR1 and BCRP.

### 2.12. Data Analysis

The transporter-mediated uptake and ATP-dependent transport were determined by subtracting the activity of the control cell/transporter abundant-membrane in the presence of AMP from the activity of the transporter-overexpressing cell/transporter-abundant membrane in the presence of ATP. The relative transporter-specific accumulation was expressed as the percent of DMSO activity. The results were presented as the mean ± SEM from 3 parallels. More than a 20% reduction in transport activity was considered as inhibition. For the IC_50_ determination, dose–response curves were fitted using a 4-parameter nonlinear model in GraphPad Prism version 9.0. Confidence intervals (CIs) of 95% were also determined.

## 3. Results

### 3.1. Uptake Transport Inhibition by Uncaria tomentosa Extract

One-point uptake inhibition studies revealed an inhibitory effect of UT extract on several uptake transporters. OCTN1, MATE1, MATE2K, ASCT2, and THTR1 were inhibited weakly, i.e., less than 50% (Figure 1).

For those SLC transporters where more than 50% inhibition was observed with the UT extract (OATP1B1, -1B3, -2B1, -1A2, OAT1-3, URAT1, OCT1, OCT2, OCT3, NTCP, ENT1, ENT2, ENT4, and SGLT2), IC_50_ values were determined (Table 1, Appendix A). UT extract did not inhibit ASCT1, LAT1, LAT2, and THTR2.

### 3.2. Efflux Transport Inhibition by Uncaria tomentosa Extract

UT extract inhibited all ABC transporters tested here with comparable potency (Figure 2), except for MRP5 where the extent of inhibition was around 50% at the highest used concentration (Table 2, Appendix A).

To assess the potential in vivo inhibitory effect of orally administered UT extract on the intestinally expressed MDR1 and BCRP, I_gut_/IC_50_ values were calculated (Equation (1)) according to the ICH M12 guideline for drug–drug interactions. For the calculations, 500 mg UT extract was used as an approximated dose (Appendix A). The results indicate that the potential risk of in vivo inhibition cannot be excluded since the calculated I_gut_/IC_50_ ratio is 3.6 (MDR1) and 18.8 (BCRP) times higher than the guideline cut-off [62].

### 3.3. Interaction of UT Active Ingredients with SLCs and ABCs

The active substances found in the largest amount in the total UT extract used here were investigated to determine whether they are responsible for the observed transporter interactions. Uncarine D, isopteropodine, and mitraphylline, the three bioactive oxindole alkaloids from the POA group, were tested (Figure 3 and Figure 4).

Similar to the studies with UT extract, transporters where the extent of inhibition exceeded 50% were selected for the follow-up IC_50_ determination. The most potent inhibitor was isopteropodine for all tested SLCs and ABCs except ENT1. An intermediary inhibitory effect of 20–50% was observed with uncarine D on OCTN1, OCT1, OCT2, MDR1, and MRP2, and with mitraphylline on OATP1A2, OCT1, OCT2, OCTN1, ENT1, ENT2, ENT4, BCRP, and MDR1. IC_50_ values are summarized in Table 3 and Table 4.

Applying the isopteropodine and mitraphylline content of the extract (Appendix A) and the theoretical dose of 500 mg UT extract, the I_gut_/IC_50_ ratios for both MDR1 and BCRP are less than the guideline cut-off value (10) [62], thereby the risk of clinical inhibition can be excluded in the intestine (Appendix A).

### 3.4. Antiproliferative Properties of the Tested Substances

To detect potential cell growth-inhibiting properties, the subchronic effect of the extract and its individual active ingredients on cell viability was determined using an MTT assay. Neither the extract nor the alkaloids did substantially influence the proliferation of the utilized transporter-overexpressing HEK293 cells (Appendix A). Since none of the substances elicited 50% growth inhibition at the highest applied concentration against any cell line, no IC_50_ values were calculated. The inhibitory effect of the extract against HEK293-OATP1B3 exceeded 40%, but all other values were around or less than 30%. The adherent cancer cells and the NIH/3T3 fibroblasts were similarly unaffected by any treatments. The general sensitivity of the utilized cells was evidenced by the IC_50_ values of the reference agent cisplatin. Based on these results, the prepared UT extract and the alkaloids isopteropodine, mitraphylline, and uncarine D exert no substantial action on the viability of the treated cells. This is consistent with the literature data in that the antiproliferative activities of UT were weak to moderate [63,64].

## 4. Discussion

The alternative remedy *Uncaria tomentosa* is commonly co-administered with clinically approved drugs [65], hence it would be crucial to know whether it alters their ADME properties, efficacy, and safety. Despite this, only sparse studies have investigated the HDI of UT extract and its components, focusing mostly on interactions with CYP enzymes and some of the main efflux transporters [66,67,68,69]. Our in vitro study addresses these outstanding issues by focusing on SLC and ABC transporter-mediated HDIs of UT extract and its purified oxindole alkaloid components, mitraphylline, isopteropodine, and uncarine D. Here, we show clear inhibitory potential on several transporters for the first time and demonstrate that both the UT extract and its three major active components, mitraphylline, isopteropodine, and uncarine D, interact with several SLC and ABC transporters.

Products from UT are available in different forms such as tea, liquids, capsules, and tablets. The bioactive compound content of these marketed products is highly variable, and depends, among other things, on the habitat of the plant, the parts used for extraction, and the season of the harvest. The extraction method is also important, since aqueous, hydroalcoholic, or ethanolic extractions result in different compositions [65,70]. As these products are considered food, and not drugs, according to FDA regulations, these details are often not indicated, making it difficult to estimate the potential effects of the different marketed products. The commercially available orally administered products have a strength range of 20–500 mg and a maximum of a 1500 mg dose/day, and are usually recommended for arthritis, rheumatism, diabetes, skin and bowel inflammations, locomotor disorders, infections, and as adjunctive therapy for cancer and strengthening the immune system [40]. On the other hand, the standardized chemical composition of UT extract used in clinical experiments is set to less than 0.5% oxindole alkaloids and 8–10% carboxyl esters and used in a 250–300 mg dose [8]. Considering the indications, and the interaction potential of the UT extract, its administration as an adjuvant therapy with substrate drugs of the intestinal uptake transporters, such as OATP2B1, OCT1, OCT3, and ENT4, may alter the absorption and consequently the efficacy of the used drugs, such as the antidiabetic drug metformin [71]. Here, we performed in vitro studies to confirm these potential interactions. Among pure oxindole alkaloids, isopteropodine was the most potent with an IC_50_ of 2.8 and 16.1 µM for OCT1 and 2 and 45.4 µM for OATP2B1. OCT1- and ENT1-mediated transport was inhibited with an IC_50_ of 7.1 and 49.6 µM by mitraphylline. In addition, all three chosen alkaloids inhibited OCTN1 by about 50%. The increasing use of UT extract as a supplement may result in sufficiently high concentrations of its active ingredients in the small intestine to cause HDIs [71]. To evaluate the potential of the in vivo HDI of UT extract, we applied the calculation for intestinal drug interactions of orally administered drugs according to the ICH M12 DDI guidance [62]. Using the recommended in vivo–in vitro extrapolation approach for DDI potential evaluation, I_gut_/IC_50_ were calculated using the in vitro IC_50_ data and a 500 mg dose [72]. The I_gut_/IC_50_ value was above the recommended ICH M12 cut-off of 10 both for MDR1 (36) and for BCRP (188) for UT extract. These results suggest a potential HDI that can cause decreased drug excretion and the increased plasma concentration of the drug. However, in the used UT extract, I_gut_/IC_50_ ratios were <10 for the three POAs both for MDR1 and BCRP, suggesting a synergistic effect of substances in UT extract on these transporters (Appendix A). On the other hand, it is important to highlight that these ratios can vary with the different UT products. These findings support the importance of the standardization of commercially available UT products for safe use. A case report described elevated plasma concentrations of atazanavir, ritonavir, and saquinavir in a patient using a concomitant UT preparation, which was explained by the fact that cat’s claw is a potent CYP3A4 inhibitor in vitro [73,74]. However, considering that these protease inhibitors are MDR1 substrates, our data suggest that MDR1 inhibition may also have contributed to this phenomenon.

It has been proven both in MDR1-overexpressing MDCKII cells and Caco-2 cells that mitraphylline is an MDR1 substrate [68]. These findings together with our data of an IC_50_ of 28.3 µM imply that mitraphylline competitively inhibits MDR1 activity. In that previous study, mitraphylline was also shown not to interact with MRPs, which is in good agreement with our results [68]. Based on our data, BCRP-mediated transport was inhibited with an IC_50_ of 10.6 µg/mL by UT extract. In contrast, in BCRP- and MDR1-overexpressing MDCKII cells using a flow cytometry assay, neither MDR1 nor BCRP interacted with UT in previous studies [45]. These findings together with our results of vesicular transport assays suggest that the MDR1 and BCRP interactor components of UT may not get into MDCKII cells, presumably due to the lack of an uptake transport system. In our assays, UT extract interacted with the liver-specific uptake transporters OATP1B1 and 1B3 with an IC_50_ of 27.5 µg/mL and 16.9 µg/mL. However, in a published study, TOA-free UT extract weakly inhibited the 8-fluorescein-cAMP accumulation in OATP1B1- and 1B3-overexpressing HEK293 cells while POA-free UT extract had a more potent effect on them [45]. In our study, OATP1B1 activity was reduced to 56% in the presence of the POA isopteropodine at 100 µM, while an IC_50_ of 24.6 µM was determined for OATP1B3, suggesting that other substances in UT extract also have an inhibitory effect on uptake activity [75].

As for the potential interaction in the kidney, a case report of cat’s claw-induced acute allergic interstitial nephritis described a patient with systemic lupus erythematosus and worsening renal function after using cat’s claw [76]. Herbal supplements similar to drugs may be toxic through multiple common pathogenic mechanisms. Of the kidney-specific transporters, UT extract inhibited SGLT2 with 72.9 µg/mL, URAT1 with 15.7 µg/mL, and OAT1-3 with 88.0, 36.3, and 25.1 µg/mL of IC_50_. Furthermore, of the examined oxindole alkaloids, only isopteropodine decreased the transport mediated by SGLT2, OAT1, and OAT2 by around 50% at 100 µM, while OAT3 was inhibited with an IC_50_ of 46.5 µM. Since renal proximal tubules are involved in active transport and urinary concentration, the local concentration of substances can be relatively high in the tubules. It is well known that clinically relevant changes in the clearance of therapeutics can occur when OAT1/3 transporter activity is inhibited. Taking all this into account, the potential renal HDI of UT extract cannot be excluded. Both UT extract and isopteropodine inhibited SGLT2, which is the target of the antidiabetic drugs canagliflozin, dapagliflozin, and empagliflozin. The fact that UT has been proposed as an auxiliary treatment for patients with type 2 diabetes mellitus (T2D) [77] raises concerns about the concomitant use of UT and the listed drugs. Based on recent reports, the FDA has revised the warnings on the drug labels of both canaglifozin and dapaglifozin to include information about acute kidney injury, and added recommendations to minimize this risk [78]. It certainly is possible that SGLT2 inhibitors may predispose patients to acute kidney injury by contributing to volume depletion. In addition, the volume and intrarenal hemodynamic effects of SGLT2 inhibitors may be synergistic when combined with frequently prescribed diuretics in this population of patients with T2D. It also mentioned that UT has a diuretic effect, so it is contraindicated with other diuretics, as they act by the same mechanism and thus increase the risk of electrolyte imbalance [65].

Regarding the potential effects on the brain, previously published studies on mice and rats implied that UT has a neuroprotective effect, while several published clinical tests showed that UT-supplemented treatments positively influence memory and cognitive functions [36,79]. Mitraphylline, isopteropodine, and UT extract inhibited the MDR1-mediated transport to a similar extent (IC_50_: 28–54 µM), and not mitraphylline but isopteropodine and UT extract inhibited BCRP with an IC_50_ of 54.6 µM and 10.6 µg/mL. Furthermore, MRP2 and MRP4 were also inhibited by the tested oxindole alkaloids and the UT extract. These ABCs are highly expressed in the BBB, reducing the penetration of their substrates into the brain. These transporters, together with the uptake transporters including OATP2B1, ASCTs, and ENT1, may play a role in the uptake of the neuroprotective phytochemicals, or they may support the penetration of effective compounds by inhibiting the efflux transporters.

Although cat’s claw has been used for different diseases, there are no unequivocal clinical results or reported pharmacokinetic data, making it difficult to evaluate the in vivo potential of transporter inhibitions in the BBB. Data are available only in mice, where the bioavailability of six orally administered *Uncaria* alkaloids (5 mg/kg) ranged between 27.3% and 68.9% and reached a maximum plasma concentration of (*C*_max_) between 305.3 ± 68.8 ng/mL and 524.5 ± 124.5 ng/mL [80].

## 5. Conclusions

In conclusion, similarly to transporter-mediated DDIs, of which in vitro investigation is required by the FDA, HDIs can also be relevant in the potential alteration in toxicity, efficacy, and tissue distribution of the examined drug. The increasing interest in alternative and traditional medical therapies that can be used as supplementary to the functional medical treatment makes it more important. According to the FDA Adverse Event Reporting System database, two adverse events associated with cat’s claw have been reported. Both cases were believed to result from the co-consumption of UT with other drugs [81]. These cases highlight the importance of the reliable prediction of clinically relevant HDIs, which in addition to in vitro HDI studies would also require the better understanding of pharmacokinetic properties of herbs including intestinal and plasma concentrations.

## Figures and Tables

**Figure 1 pharmaceutics-16-01363-f001:**
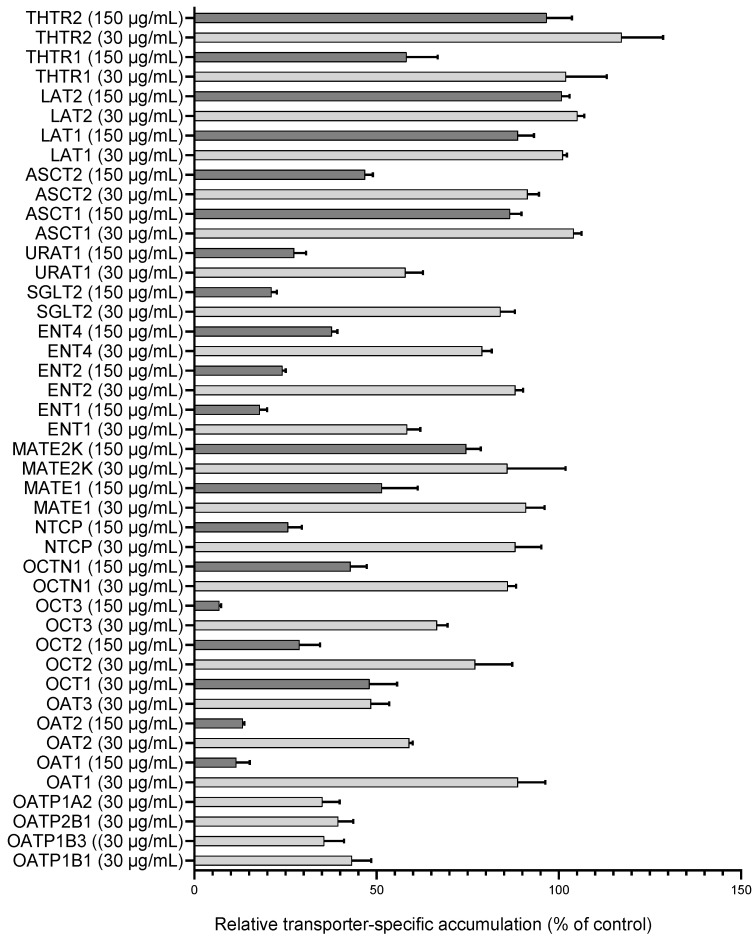
One/two-point uptake inhibition with UT extract. Relative transporter-specific transport of the validated transporter substrates in the presence of 30 or 150 µg/mL UT extract. Data are relative to the vehicle control. All data are presented as the mean ± SEM.

**Figure 2 pharmaceutics-16-01363-f002:**
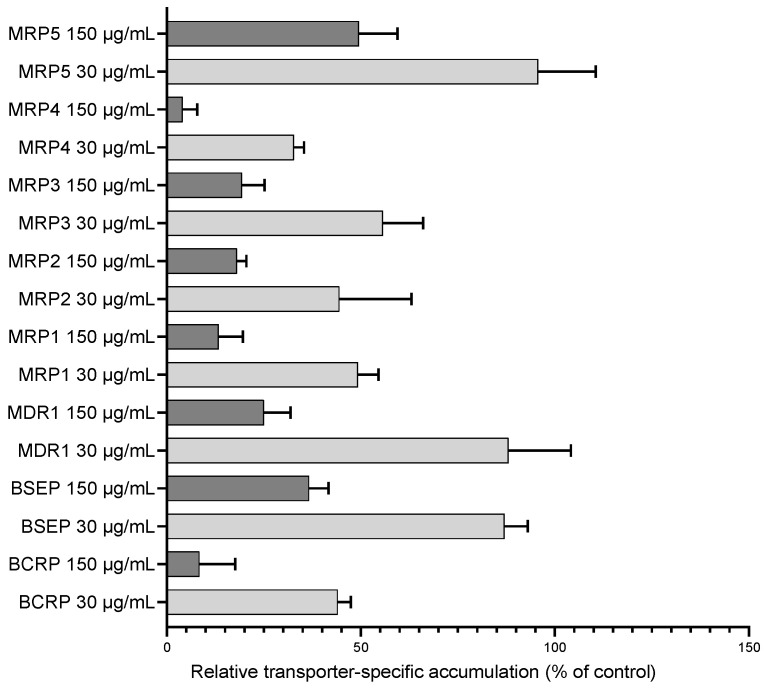
One/two-point inhibition of ABC transporters with UT extract. Relative ATP-dependent transport of the validated transporter substrates in the presence of UT extract. Data are relative to the vehicle control. All data are presented as the mean ± SEM.

**Figure 3 pharmaceutics-16-01363-f003:**
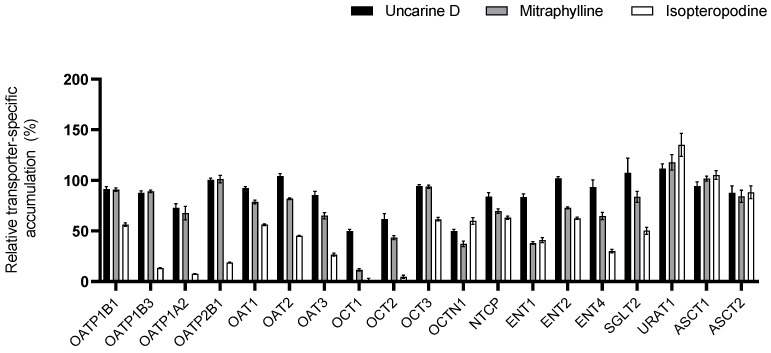
Relative transporter-specific transport of the validated SLC transporter substrates in the presence of 100 µM uncarine D, mitraphylline, or isopteropodine. Data are relative to the vehicle control. All data are presented as the mean ± SEM.

**Figure 4 pharmaceutics-16-01363-f004:**
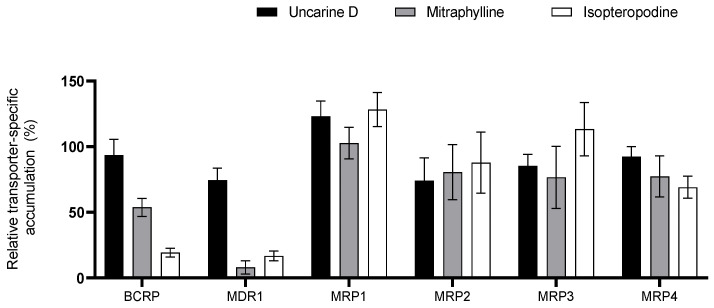
Relative transporter-specific transport of the validated ABC transporter substrates in the presence of 100 µM uncarine D, mitraphylline, or isopteropodine. Data are relative to the vehicle control. All data are presented as the mean ± SEM.

**Table 1 pharmaceutics-16-01363-t001:** IC_50_ values determined in transporter-overexpressing HEK293 or MDCKII cell lines using UT extract.

Transporter	IC_50_ (µg/mL) (95% CI of IC_50_)
OATP1B1	27.56 (22.47 to 38.06)
OATP1B3	16.95 (13.56 to 21.30)
OATP2B1	40.92 (36.89 to 46.50)
OATP1A2	45.15 (28.05 to 186.5)
OAT1	88.03 (64.56 to 213.7)
OAT2	36.34 (30.45 to 48.15)
OAT3	25.11 (19.21 to 39.18)
URAT1	15.66 (13.36 to 18.06)
OCT1	29.66 (14.61 to 45.59)
OCT2	65.33 (45.70 to 95.38)
OCT3	60.16 (45.94 to 76.47)
NTCP	54.32 (33.45 to 58.54)
ENT1	13.22 (4.79 to 20.51)
ENT2	107.1 (70.13 to 187.9)
ENT4	95.50 (83.61 to 109.4)
SGLT2	72.89 (65.01 to 81.51)

**Table 2 pharmaceutics-16-01363-t002:** IC_50_ values of UT extract determined for efflux transporters.

Transporter	IC_50_ (µg/mL) (95% CI of IC_50_)
BCRP	10.63 (6.265 to 56.17)
BSEP	105.0 (53.73 to 201.3)
MDR1	54.56 (27.87 to 76.74)
MRP1	21.82 (15.86 to 30.40)
MRP2	59.78 (37.40 to 84.75)
MRP3	48.54 (30.77 to 84.30)
MRP4	13.50 (10.60 to 16.93)

**Table 3 pharmaceutics-16-01363-t003:** IC_50_ values of mitraphylline and isopteropodine determined in transporter-overexpressing cell lines.

	IC_50_ (µM) (95% CI of IC_50_)	
Transporter	Mitraphylline	Isopteropodine
OATP1B1	-	-
OATP1B3	-	24.57 (11.84 to 36.98)
OATP2B1	-	45.44 (43.63 to 47.29)
OATP1A2	-	36.05 (33.62 to 38.59)
OAT3	-	46.52 (39.61 to 53.84)
OCT1	7.05 (6.571 to 7.566)	2.81 (2.392 to 3.296)
OCT2	-	16.10 (13.48 to 20.07)
SGLT2	-	72.69 (57.89 to 90.86)
ENT1	49.59 (35.79 to 67.38)	-

**Table 4 pharmaceutics-16-01363-t004:** IC_50_ values of mitraphylline and isopteropodine determined for BCRP and MDR1.

	IC_50_ (µM) (95% CI of IC_50_)	
Transporter	Mitraphylline	Isopteropodine
BCRP	-	54.61 (41.54 to 74.74)
MDR1	28.34 (21.67 to 37.08)	28.25 (17.95 to 45.90)

## Data Availability

The data presented in this study are available on request from the corresponding author. The data are not publicly available due to the company’s participation.

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
