# Peer review of "Modulation of Multispecific Transporters by Uncaria tomentosa Extract and Its Major Phytoconstituents"

_pharmaceutics, 2024, doi:10.3390/pharmaceutics16111363_

Round 1
Reviewer 1 Report
Comments and Suggestions for Authors
This very interesting study investigating the effect of Uncaria tomentosa extract, as well as selected alkaloids present in this extract, on the numerous targeted protein transporters affecting their activity and pointing possible drug-plant extract interactions.
The following aspects of the manuscript should be addressed:
1) Latin names of plant species and genera should be written using Italic front;
2) Introduction:
Line 53: “….herbal medicinal products are on..” – too general statement in given context;
Lines 106-107: Uncaria tomentosa is not a compound;
Lines 110-112: “This study focuses on the in vitro investigation of UT extract and its selected bioactive compounds induced HDI mediated by a broad set of relevant SLC, and ABC transporters.” – unclear, please specify what was the aim?
3) Materials and Methods:
The inner bark of stem or root was used?
If “the sum of mitraphylline and uncarine D was determined” how the validation was performed? For mixture of standard compounds?
“From the extract three sample preparations were made, and each sample was injected in triplicate.” – what for?
“4.265 ± 0.1936 mg in 1 g” – just 4.265 ± 0.1936 mg/g of extract and so on;
Line 155: where standard compounds were purchased?
Line 208: which alkaloids? Much more details should be given;
Line 219: which alkaloids? Much more details should be given;
Line 236: which tested substances?
4) Results:
Lines 271-273: this text should be removed;
Figure 1, Figure 2: it would be beneficial for these figures if different doses are presented in different colours;
and also for Figure 3, Figure 4:“of three parallels ± SEM” - ?
Line 285, 29This very interesting study investigating the effect of Uncaria tomentosa extract, as well as selected alkaloids present in this extract, on the numerous targeted protein transporters affecting their activity and pointing possible drug-plant extract interactions.
The following aspects of the manuscript should be addressed:
1) Latin names of plant species and genera should be written using Italic front;
2) Introduction:
Line 53: “….herbal medicinal products are on..” – too general statement in given context;
Lines 106-107: Uncaria tomentosa is not a compound;
Lines 110-112: “This study focuses on the in vitro investigation of UT extract and its selected bioactive compounds induced HDI mediated by a broad set of relevant SLC, and ABC transporters.” – unclear, please specify what was the aim?
3) Materials and Methods:
The inner bark of stem or root was used?
If “the sum of mitraphylline and uncarine D was determined” how the validation was performed? For mixture of standard compounds?
“From the extract three sample preparations were made, and each sample was injected in triplicate.” – what for?
“4.265 ± 0.1936 mg in 1 g” – just 4.265 ± 0.1936 mg/g of extract and so on;
Line 155: where standard compounds were purchased?
Line 208: which alkaloids? Much more details should be given;
Line 219: which alkaloids? Much more details should be given;
Line 236: which tested substances?
4) Results:
Lines 271-273: this text should be removed;
Figure 1, Figure 2: it would be beneficial for these figures if different doses are presented in different colours;
and also for Figure 3, Figure 4:“of three parallels ± SEM” - ?
Line 285, 298, 339: “(Error! Reference source not 285 found.” - ?
5) Discussion:
433: “In our hand...” -? 285, 339: “(Error! Reference source not 285 found.” - ?
5) Discussion:
433: “In our hand...” -?
Reviewer 2 Report
Comments and Suggestions for Authors
Major:
1. I believe it is necessary to know how many of the interactions have been described in the literature in a systematic way and not just manually checking (Supplementary Table). For that, I suggest using the data from https://jcheminf.biomedcentral.com/articles/10.1186/s13321-023-00778-w (https://github.com/enveda/ethnobotany/raw/main/data/processed/plant_chemical_associations.tsv.zip) and finding all the compounds of the herb investigated here. Next, you can compare them against the ones you found and how many of them are targeting the proteins investigated according to ChEMBL. That way you can claim that some of your findings are novel and have not been described before.
2. Add a reference to the first sentence in the introduction
3. "Most of the pharmacological studies has focused 46
on the anti-inflammatory and immunomodulatory activity of the plant [10]–[14]" confirm that this is the case with (PMID: 37701812), which contains all the uses of this plant described in the literature (https://github.com/enveda/ethnobotany/raw/main/data/processed/plant_disease_associations.tsv.gz) and give a ratio of how many of the total diseases are anti-inflammatory and immunomodulatory to back up your claims.
4. Data should be released in zenodo in a computable format instead of tables in the PDF.
Comments on the Quality of English Language
None
Round 2
Reviewer 2 Report
Comments and Suggestions for Authors
Data should be released in zenodo in a computable format instead of tables in the PDF. Our study was funded by a Hungarian grant and not funded by EU therefore it is not required to share data in Zenodo. However, we are happy to share all data upon request.
This is not acceptable sorry but data has to be shared to guarantee reproducibility since it does not contain any clinical or sensible data.
Also some of the references are not pointing to the articles but to GitHub/online links.
